# Folinate Supplementation Ameliorates Methotrexate Induced Mitochondrial Formate Depletion In Vitro and In Vivo

**DOI:** 10.3390/ijms22031350

**Published:** 2021-01-29

**Authors:** Nga-Lai Sou, Yu-Hsuan Huang, Der-Yuan Chen, Yi-Ming Chen, Feng-Yao Tang, Hsin-An Ko, Yi-Hsuan Fan, Yi-Ying Lin, Yi-Cheng Wang, Hui-Ming Chih, Barry Shane, Wen-Nan Huang, En-Pei Isabel Chiang

**Affiliations:** 1Food Science and Biotechnology, National Chung Hsing University (NCHU), Taichung 402, Taiwan; looksusan2013@gmail.com (N.-L.S.); jumptone@hotmail.com (Y.-H.H.); inhsinkuo@gmail.com (H.-A.K.); imy-0808@yahoo.com.tw (Y.-H.F.); yoy610287@hotmail.com.tw (Y.-Y.L.); jjqqkk6688@gmail.com (Y.-C.W.); fina3589@gmail.com (H.-M.C.); 2Innovation and Development Center of Sustainable Agriculture (IDCSA), National Chung Hsing University (NCHU), Taichung 402, Taiwan; 3Allergy Immunology Rheumatology, Taichung Veterans General Hospital (TVGH), Taichung 402, Taiwan; dychen1957@mail.cmu.edu.tw (D.-Y.C.); ymchen1@vghtc.gov.tw (Y.-M.C.); wennanhuang215@gmail.com (W.-N.H.); 4Allergy Immunology Rheumatology, China Medical University Hospital, Taichung 402, Taiwan; 5Department of Nutrition, China Medical University, Taichung 402, Taiwan; vincenttang@mail.cmu.edu.tw; 6Department of Nursing and Pediatrics, Taichung Veterans General Hospital (TVGH), Taichung 402, Taiwan; 7Nutritional Sciences and Toxicology, UC Berkeley, Berkeley, CA 94701, USA; bandie@berkeley.edu

**Keywords:** metabolism, methotrexate, therapeutic drug monitoring, folinate, one carbon metabolism, rheumatoid arthritis, formate, immunosuppressants

## Abstract

(1) Background: Antifolate methotrexate (MTX) is the most common disease-modifying antirheumatic drug (DMARD) for treating human rheumatoid arthritis (RA). The mitochondrial-produced formate is essential for folate-mediated one carbon (1C) metabolism. The impacts of MTX on formate homeostasis in unknown, and rigorously controlled kinetic studies can greatly help in this regard. (2) Methods: Combining animal model (8-week old female C57BL/6JNarl mice, *n* = 18), cell models, stable isotopic tracer studies with gas chromatography/mass spectrometry (GC/MS) platforms, we systematically investigated how MTX interferes with the partitioning of mitochondrial and cytosolic formate metabolism. (3) Results: MTX significantly reduced de novo deoxythymidylate (dTMP) and methionine biosyntheses from mitochondrial-derived formate in cells, mouse liver, and bone marrow, supporting our postulation that MTX depletes mitochondrial 1C supply. Furthermore, MTX inhibited formate generation from mitochondria glycine cleavage system (GCS) both in vitro and in vivo. Folinate selectively rescued 1C metabolic pathways in a tissue-, cellular compartment-, and pathway-specific manner: folinate effectively reversed the inhibition of mitochondrial formate-dependent 1C metabolism in mouse bone marrow (dTMP, methionine, and GCS) and cells (dTMP and GCS) but not methionine synthesis in liver/liver-derived cells. Folinate failed to fully recover hepatic mitochondrial-formate utilization for methionine synthesis, suggesting that the efficacy of clinical folinate rescue in MTX therapy on hepatic methionine metabolism is poor. (4) Conclusion: Conducting studies in mouse and cell models, we demonstrate novel findings that MTX specifically depletes mitochondrial 1C supply that can be ameliorated by folinate supplementation except for hepatic transmethylation. These results imply that clinical use of low-dose MTX may particularly impede 1C metabolism via depletion of mitochondrial formate. The MTX induced systematic and tissue-specific formate depletion needs to be addressed more carefully, and the efficacy of folinate with respect to protecting against such depletion deserves to be evaluated in medical practice.

## 1. Introduction

Formate plays a key role in mammals’ cellular and whole-body metabolism. The mitochondrial production of formate is a major process for the endogenous generation of folate-related one-carbon (1C) moieties [1]. Formate is required for the biosynthesis of nucleotides [2], free adenosine pool, methionine, *S*-adenosylmethionine (adoMet) [3]. Formate also protects the cytosolic pool of folate cofactors [4] and is used to re-synthesize serine via cytosolic 1C metabolism [5]. In cells with defective mitochondrial 1C metabolism, the cytosolic pathway can compensate for the loss of mitochondrial formate production. When both the cytosolic and mitochondrial pathways are compromised, cells need exogenous formate or endogenous formaldehyde [6] as alternative 1C sources.

Low-dose methotrexate (MTX) is one of the most commonly used immunosuppressants in rheumatic and other inflammatory conditions. We proposed that MTX interferes with formate metabolism in numerous ways. First, MTX may interfere in formate production by redistributing folate cofactors. MTX functions by inhibiting dihydrofolate reductase (DHFR) and thymidylate synthase (TS), and inhibition of these enzymes adversely affects the survival of rapidly replicating cells by suppressing de novo synthesis of nucleotides and folate-dependent reactions. In mammals, 10-formyl tetrahydrofolate (THF) dehydrogenase effectively burns 1C units and therefore contributes to the turnover of formate [3]. 10-formyl-THF is converted to formate and THF by mitochondrial monofunctional 10-formyl-THF synthetase (MTHFD1L). In MTX-treated rat liver, there was a relative decrease in 5-methyl THF and an increase in formylated THF [7]. As formylated THF serves as an important source of formate, it is plausible that MTX-induced accumulation of 10-formyl THF disturbs formate homeostasis.

Second, MTX may impede formate generation via alterations of the serine and glycine supply. Approximately 50% of plasma formate is derived from serine, and serine starvation reduces formate synthesis in vivo [8]. Glycine is a secondary contribution to formate in mammals [3]. In MTX-treated MCF-7 cells, the uptake of serine and glycine from the media decreases upon MTX treatment in proportion to the decrease in the serine and glycine consumption rate for protein synthesis. The net rate of serine to glycine conversion also decreases significantly in these cells [9].

Third, MTX may inhibit formate production via the inhibition of the mitochondrial respiratory chain. Disruption of the respiratory chain impairs mitochondrial production of formate in isolated mitochondria of rat liver [10]. Respiratory chain inhibition due to serine withdrawal in T-REx-293 cells can be rescued by purine or formate supplementation [11]. By oxidizing NADH to NAD+ in the mitochondrial matrix, Complex I (NADH: ubiquinone oxidoreductase) reduces equivalents for many reactions, including the TCA cycle. The mitochondrial NAD kinase catalyzes the phosphorylation of NAD to yield NADP. MTX has been shown to inhibit the catalytic activity of mitochondrial NAD kinase (NADK2) and numerous NAD(P)-dehydrogenases in hepatocytes [12] and mouse livers [13]. MTX decreases the activity of mitochondrial respiratory complexes, inhibits oxygen consumption, impairs mitochondrial oxidative phosphorylation and ATP synthesis in rodent liver and intestine [14,15].

Based on this evidence, we postulated that long-term clinical use of MTX might lead to impaired formate homeostasis. Whether MTX depletes formate pools and turnover and the efficacy of clinical folinate rescue on formate homeostasis during MTX therapy is unknown. Considering that long-term and multiple medications use, the complexity and multiple factors in human subjects that may have masked the impact of MTX on formate. Combining animal and cell models, we systematically explored whether clinical use of low-dose MTX may cause formate depletion and investigated the consequences and therapeutic approaches for dealing with abnormal formate metabolism.

## 2. Results

### 2.1. MTX Inhibited Endogenous and Promoted Exogenous Formate Utilization for Nucleotide and Methionine Biosyntheses In Vitro

MTX decreased the M+1 and M+2 enrichments in purines from L-[3-^13^C]serine by >90% and decreased the relative enrichments (ratio of dA, dG to Ser+1 [16,17]) by ~56% (Table 1A,B). The inhibition on dTMP enrichments (dT+1) were less (by 56%, *p* = 0.004). There is a mild but significant increase in the exogenous formate utilization for dTMP synthesis (by +3.4%, *p* = 0.017) (Table 1C). Taken together, MTX inhibited the incorporation and utilization of endogenous formate generated from L-[3-^13^C]serine, and increased the utilization of 1C moiety from exogenous [^13^C]formate for nucleotide synthesis (Figure 1a,b).

MTX inhibited endogenous and promoted exogenous formate utilization for methionine biosynthesis (Table 2A,C) in vitro. The partitioning of 5,10 methyleneTHF between dTMP and methionine synthesis was calculated [18] (Table 2B,D). The inhibition of endogenous formate utilization was stronger in methionine synthesis compared to that of dTMP. Consistently, more exogenous formate was utilized for methionine synthesis during MTX treatment (enrichments increased by 110%).

### 2.2. MTX Interferes with the Conversion between Serine and Glycine

MTX or folinate did not alter the sum of all serine isotopic species (Ser M+1+2+3) in either cell or mouse tissues examined, but both significantly altered the distributions among the isomers in serine and glycine (Table 3, Figure 2a,b). MTX inhibited glycine synthesis. In the cell model, MTX decreased enrichments from L-[2,3,3-^2^H_3_]serine in Gly+1, Ser+1, and Ser+2 by 23%, 63%, and 26%, respectively. Reductions in Ser+1 were observed in the cell model, in the mouse liver, and bone marrow, consistent with the decreased mitochondrial formate generation. Only co-treatment with folinate more effectively ameliorated the MTX-inhibition in Ser+1 in cells. In the mouse bone marrow, MTX inhibited Ser+1, Ser+3, and Gly+1 but not Ser+2. MTX decreased Ser+1 and increased Ser+2 and Ser+3 in the liver. The increased Ser+2 may indicate that more cytosolic 5,10-CD_2_THF was used for serine synthesis during MTX treatment. Folinate could not restore Gly+1 in the cells, mouse liver, or bone marrow. Folinate restored the Ser+1 and Ser+3 enrichments to a normal level only in the liver but not in the marrow, suggesting a significant tissue difference in response to folinate rescue with respect to 1C supply (Figure 2a,b).

### 2.3. The Partitioning of 1C Metabolic Fuxes via MethyleneTHF between Mitochondria- and Cytosolic-Derived Formate

The partitioning between the dTMP and methionine synthetic pathways was calculated [17] and compared in cells (Table 2B,D) and in vivo (Table 4C). Under MTX treatment, dTMP had a higher priority than methionine synthesis using endogenous formate derived from serine. The partitioning of 1C metabolic fluxes between mitochondrial and cytosol also differed between the liver and bone marrow. Table 3 and Table 4A shows the utilization of serine for glycine, methionine, and dTMP synthesis in the bone marrow and liver proteins. The results of the current study were consistent with those findings [17,20] that approximately 90% of dTMP synthesis is from mitochondria formate. MTX strongly reduced dTMP (dT+1) and methionine (Met+1) enrichments from mitochondria-derived formate, but not from cytosolic methyleneTHF via cSHMT (dT+2 and Met+2) in cells or mouse tissues (Figure 3a–d, Table 4B). In fact, the increased enrichments in dT+2 may indicate a significant role of cSHMT in supplying 1C unit for synthesis during MTX treatment. MTX inhibited mitochondrial formate utilization for dTMP synthesis (from 93% to 66%), which was restored to normal by folinate in the marrow (88%) and liver (Control 88%; MTX 73%; MTX+FA 88%). MTX inhibited mitochondrial formate utilization for methionine synthesis (from 74% to 65%) that could also be restored by folinate to the normal level in the marrow (71%); however, folinate failed to rescue methionine synthesis from mitochondrial formate utilization in the liver (control 63%; MTX 34%; MTX+FA 36%) (Figure 3b, Table 4B).

Taken together, MTX inhibited de novo dTMP and methionine synthesis from mitochondria-derived formate but not from serine through cSHMT. These results demonstrated significant tissue specificity of the utilization of serine-derived 1C moiety and methyleneTHF between the mitochondria and cytosol during MTX treatment and folinate rescue (Figure 3d).

### 2.4. Folinate Supplementation Rescued the MTX-Inhibited De Novo Thymidylate Biosynthesis from Glycine via GCS

We recently established a feasible approach to trace the utilization of 1C moiety from mitochondrial GCS using [2-^13^C]glycine [21]. Labeled glycine at carbon two has been given to humans to investigate its direct incorporation into purines and excretion as uric acid [22] rather than its metabolic route via the mitochondria GCS. In our approaches, enrichments of specific metabolite isomers were selected for above purpose. Particularly, since the carbon two of glycine directly incorporates into serine as Ser+1 but not dTMP or methionine in the cytosol, the dT+1, Ser+2, Met+1, and mC can only be observed when the mitochondrial GCS derived formate from the 2-carbon of [2-^13^C]glycine enters the cytosol, incorporates into methyleneTHF and used for cytosolic serine, dTMP, methionine, and mC synthesis. This enables us to trace how MTX and folinate change the utilization of GCS-produced 1C moiety in vitro and in vivo.

MTX decreased Gly+1 by 18% and Ser+1 by 42% in cells (Table 5A); MTX decreased Gly+1 by 34% without changing Ser+1 in the marrow (Table 5B). Folinate supplementation fully restored Ser+1 and Gly+1 from [2-^13^C]glycine in cells, mouse bone marrow, and the liver. Interestingly, folinate supplementation significantly increased Gly+1 and Ser+1 in the mouse bone marrow and Gly+1 in the liver compared to the controls. MTX inhibited dA+1 and dG+1 from [2-^13^C]glycine by 46% and 36% in cells (Table 5A); MTX inhibited dA+1 and dG+1 by 13% and 16% in the mouse marrow (Table 5C). Folinate alone did not alter glycine incorporation in purines in vitro or in vivo. Folinate supplementation in MTX treatment successfully restored glycine incorporation in purines in vitro (Table 5A) and in vivo (Table 5C).

With respect to GCS-derived formate in dTMP synthesis and the transmethylation pathway, MTX significantly decreased dT+1 in cells by 64% (*p* = 0.044) (Table 5A) and in the mouse marrow by 12% (*p* = 0.024) (Table 5C). MTX significantly inhibited the incorporation of [2-^13^C]glycine into mC via mitochondrial GCS in mouse marrow by −44%. Finally, folinate supplementation fully restored de novo dTMP biosynthesis via mitochondria GCS in both the cell model and the mouse bone marrow (Table 5A,C). In summary, MTX inhibited the utilization of GCS-derived formate, and folinate is effective in rescuing certain impaired 1C metabolic pathways (Figure 4).

## 3. Discussion

### 3.1. Low-Dose MTX Depletes Mitochondrial Formate Production

The present study demonstrated that clinical use of low-dose MTX particularly impedes 1C metabolism via depletion of mitochondrial formate in vitro and in vivo. Formate is produced in folate-mediated pathways in the mitochondria (from serine, glycine, sarcosine, and dimethylglycine) and cytoplasm (from serine and histidine), and also from folate independent sources including methanol, tryptophan, α-oxidation of branched fatty acids, or cholesterol synthesis [23]. Nevertheless, we clearly demonstrated that MTX and folinate drastically change the partitioning of 1C metabolic fluxes in vivo. Mitochondria isolated from SHMT2 mutant CHO cell glyA exhibited a 94% reduction in dTMP synthesis capacity [24]. As SHMT2 accounts for the majority of 1C supply for the dTMP synthesis capacity, long-term clinical MTX treatment in humans may lead to systematic and tissue-specific formate depletion that needs to be addressed more carefully; and the efficacy of folinate with respect to protecting against such depletion deserves to be carefully evaluated.

### 3.2. Low-Dose MTX Changes the Partitioning between Mitochondrial and Cytosolic 1C Metabolic Fluxes

Most proliferating mammalian cell lines use the mitochondrial pathway as the default source for supplying 1C units [5]; under conditions when the mitochondrial pathway is disturbed, the cytosolic pathway may be activated to serve as the alternative 1C source that compensates for the demand of 1C supply. We did observe a cytosolic 1C compensation, but such compensation can only partially rescue the perturbed 1C metabolism in hepatic transmethylation (Figure 3a–c). The total 1C fluxes (mitochondrial and cytosolic) entering the folate-dependent methionine synthesis were drastically reduced by MTX in spite of the compensatory cytosolic 1C supply, and the supply of cytosolic 1C to different folate mediated nucleotide and methylation reactions appeared to have different metabolic priority among folate-mediated pathways.

### 3.3. Mitochondria Formate Supply for Folate-Mediated dTMP and Methionine Synthesis in Bone Marrow and Liver

Beta-carbon of serine is the main source for folate-mediated 1C metabolism. Compared to bone marrow, a much smaller proportion of serine tracer was incorporated into dTMP in the liver. In the liver, more 1C moieties were used in folate-dependent homocysteine remethylation for methionine synthesis compared to that for dTMP. On the other hand, while mitochondria-derived formate accounted for 88–93% of the 1C used for dTMP synthesis, it accounted for less in the methionine biosynthesis (80% in cell; 63%, in liver; 74% in marrow) (Figure 3a,b). Folate-mediated methionine synthesis used relatively less 1C from mitochondria and more 1C from cytosol compared to that of dTMP.

### 3.4. Inhibition of 1C Supply by MTX Is Specific to Tissue, Pathway, and Cellular Compartments

Folate-mediated 1C metabolism has a different metabolic priority among different tissues [25]. Transmethylation and DNA methylation is highly complex and varies among cell types [26,27]. It was reported that 5-methylTHF accounted for about 25% of the endogenous folates in rat bone marrow and about 18% of those in rat liver [28]. These differences may account for the varied responses to MTX and folinate. Indeed, bone marrow had more dTMP and methionine enrichments from serine than those in the liver; presumably because bone marrow is more proliferative than the liver. We also discovered different metabolic priorities among tissues in MTX/folinate treatment with respect to dTMP and methionine synthesis from the mitochondrial-derived formate. The MTX inhibition of dTMP synthesis from mitochondrial-derived formate (dT+1) was stronger in the bone marrow (from 93% to 66%, reduced by 27%) compared to that in the liver (from 88% to 73%, reduced by 15%). In contrast, the inhibition of methionine synthesis from mitochondrial-derived formate (Met+1) was much stronger in the liver (from 63% to 34%, reduced by 29%) compared to that in the bone marrow (from 74% to 65%, reduced by 9%) (Figure 3a,b). These data suggest specificity in tissue, pathway, and cellular compartment with respect to the MTX inhibition on mitochondrial 1C supply. These findings provide insights on how to correct MTX induced biochemical alterations by folinate rescue or other approaches.

### 3.5. MTX Inhibits Formate Supply from the Mitochondria Glycine Cleavage System

The impacts of MTX on GCS, another source of mitochondrial formate [3], has never been investigated before. In the present study, we detected the incorporations of the [2-^13^C]glycine carbon in dTMP and methyl-cytosine in the mouse bone marrow, and dTMP and methionine in the cell model. During glycine decarboxylation, the GCS-derived 1C moiety is exported from the mitochondria as formate, and the original ^13^C-labeled glycine 2-carbon is transferred to THF to yield ^13^C-labeled methyleneTHF, which ultimately produces ^13^C-labeled formate. Theoretically, this 1C unit can be utilized in cytosolic biosynthesis for dTMP, serine, methionine, methyl-cytosine. Our study demonstrated the novel finding that low-dose MTX significantly inhibited formate generation from GCS (Table 5). The clinical significance of these findings is currently under investigation.

### 3.6. Folinate Rescues 1C Supply for Nucleotides but Failed to Rescue Hepatic Transmethylation During MTX Therapy

Purine synthetic pathways have different metabolic roles among tissues. The purine pathway in the liver is more linked to the turnover of purine nucleotides in intermediary metabolism (NADH, ATP, and FAD) and for RNA; in the marrow, it is more linked to the requirement and supply of adenine and guanine for DNA and RNA synthesis [29]. The activity of purine synthetic enzyme aminoimidazole carboxamide ribonucleotide transformylase (AICART) and glycinamide ribonucleotide transformylase (GART) was several times greater in the marrow than in the liver [29], indicating that bone marrow is the active site of purine synthesis. Inhibition of GART in rats exposed to nitrous oxide was prevented by supplying methylthioadenosine, a formate precursor. These data suggest that formate supply or even folinate might be effective in rescuing purine synthesis in vivo. In fact, we discovered that the strong inhibition in purine synthesis could be fully rescued by folinate in the mouse bone marrow during MTX therapy.

Furthermore, folinate supplementation successfully rescued the MTX-inhibited de novo dTMP biosynthesis from the mitochondria and further reversed the 1C flow from the cytosol used to compensate for dTMP synthesis in a cell model, mouse liver, and bone marrow. Suppression in total dTMP enrichments (dT+1 and dT+2) was fully restored to the normal control level by folinate (Figure 3a–d).

In contrast to nucleotide synthesis, folinate failed to rescue 1C supply for methionine synthesis in the liver. The proportional 1C fluxes used for methionine synthesis changed from 63% to 34%, which remained unchanged by folinate supplementation (Figure 3a–d).

Taken together, folinate supplementation only partially alleviated the adverse effects of MTX. Folinate supplementation can rescue dTMP and purine syntheses in mouse bone marrow and liver, but it failed to fully recover the transmethylation carbon flow in the liver. Deacon et al. showed that more activity of methionine synthetase occurred in rat liver than in marrow [29]. The liver is the main capital for transmethylation metabolism. Low-dose MTX inhibits the key enzyme for adoMet synthesis methionine adenosyltransferase (MAT) in vitro and in vivo, and the disturbed adoMet homeostasis could not be restored by folinate when cells were pre-exposed to MTX [30]. As polyglutamated MTX stays in the hepatocytes, we raised concerns about the efficacy of clinical folinate rescue with respect to maintaining hepatic adoMet status and methylation. The transmethylation kinetics during MTX therapies certainly deserve further attention. Our present study proved that although folinate can effectively reverse the mitochondria-derived formate-dependent dTMP synthesis in bone marrow and the liver, folinate failed to assist methionine synthesis. These findings, along with the fact that MTX inhibits MAT, certainly make it difficult to restore the transmethylation machinery in patients on long-term MTX therapy. Research should be conducted to evaluative the impacts of formate and/or adoMet supplementation during MTX therapy for alleviation of its adverse effects.

## 4. Materials and Methods

### 4.1. Cell Model, Culture Condition, and Treatment Protocol

All chemicals were purchased from Sigma Chemical Company (St. Louis, MO, USA) via the local distributor in Taiwan, unless otherwise specified. Adult human hepatic cell lines L02 (kindly provided by Dr. Shuang-En Chuang at National Institute of Cancer Research, Miaoli County, Taiwan) and HepG2 cells expressing glycine-N methyltransferase GNMT (GNMT+) [19,31] were used as liver-derived cell models. L02 was reported to express adoMet synthase (MAT)1A, betaine-homocysteine methyltransferase (BHMT), and methionine synthase (MTR) but not MAT2A [32], which would be more relevant to normal hepatocyte metabolism. GNMT+ cell was chosen as we have thoroughly characterized 1C metabolic kinetics in this model [19,33,34], and it would be a better cell model for investigating hepatic metabolism compared to wildtype HepG2 cells [35].

Cells were grown in minimal essential medium (MEM) or alpha minimal essential medium (αMEM) containing 10% FBS (Gibco, Gaithersburg, MD, USA), penicillin (100,000 units/L), streptomycin (100 mg/L), amphotericin (0.25 mg/L) in 5% CO_2_ at 37 °C, and the media were replaced every 72 h. MTX dose at 50 nM was chosen based on relevant physiological concentration in human cells and on our previous studies [30]. Intracellular MTX concentration was 18–51 nM in erythrocytes of rheumatoid arthritis (RA) patients on low-dose MTX therapy [36]. HepG2+GNMT cells were treated with MTX (50 nM) with and without folinate supplementation (100 nM) before, during, or after the MTX treatment for 24 h, defined as “pre-treatment”, “co-treatment”, and “post-treatment”, respectively (Figure 5).

Cells were cultured (3 × 10^5^) with designated tracer L-[3-^13^C]serine, [^13^C]formate, L-[2,3,3-^2^H_3_]serine, or [2-^13^C]glycine (Cambridge Isotope Laboratories, Woburn, MA, USA) that were substituted for the unlabeled counterparts as needed. During the labeling period, the unlabeled amino acid in αMEM was replaced by the designated tracer, and the medium was replaced with fresh tracer-containing medium every 24 h thereafter. All cells were harvested 24 h after the final medium replacement (total labeling for 72 h). Cells were collected, extracted [27], and derivatized for further analyses as previously described [16,35]. All experiments were performed in duplicates or triplicates independently at least two times, and data from one experiment are presented here.

### 4.2. Effects of Low-Dose MTX on Endogenous and Exogenous Formate Utilization In Vitro

The beta-carbon of serine is the main 1C donor in proliferating cells. Serine enters the mitochondria, and its hydroxymethyl group is released from the mitochondria as formate. Endogenous and exogenous formate utilization was traced by L-[3-^13^C]serine [16] and [^13^C]formate [17], respectively. L02 cells cultured in MEM media were incubated with the addition of glycine, vitamin B12, and L-[3-^13^C]serine to reach a final concentration of 0.667 mM glycine, 0.001 mM B12, and 0.25 mM serine. In a parallel experiment, 0.25 mM [^13^C]formate was used instead. Enrichments in the target metabolites, including purines, deoxythymidine (dTMP), and methionine syntheses, were determined to quantify the incorporations of 1C moiety from endogenous and exogenous formate (Figure 6).

### 4.3. Partitioning between Mitochondrial and Cytosolic 1C Metabolism

More than 90% of 1C moieties used for cytoplasmic folate-dependent metabolism are derived from mitochondrial 1C metabolism as formate. GNMT+ cells [31], [19] were cultured in α-MEM medium and treated with 50 nM MTX and 100 nM folinate for 24 h. In L-[2,3,3-^2^H_3_]serine experiments, methyleneTHF supplied by cytosolic serine hydroxymethyl-transferase (cSHMT) and incorporated directly into methionine or dTMP retains the two deuterium atoms (CD_2_) on the hydroxymethyl group of serine. If the L-[2,3,3-^2^H_3_]serine enters the mitochondria and the hydroxymethyl group is released from the mitochondria as formate, it only contains a single deuterium atom (CD_1_) [20,37] (Figure 7a,b).

### 4.4. Tracing the Metabolic Fate of Mitochondria-Derived Formate from Glycine Cleavage System

During glycine decarboxylation, the glycine cleavage system (GCS)-derived 1C moiety is exported from the mitochondria as formate. Labeled glycine at 2-carbon was given to humans to investigate its direct incorporations into purines and excreted as uric acid [22], rather than investigating its metabolic route via the mitochondria GCS. In our [2-^13^C]glycine experiment, the original ^13^C-labeled glycine 2-carbon is transferred to THF to yield ^13^C-labeled methyleneTHF, that ultimately produces ^13^C-labeled formate; this 1C unit can be utilized in cytosolic biosyntheses for nucleotides, serine, methionine, adoMet, and methyl-cytosine [21].

### 4.5. Animal Study Design

All animal experiments were approved by the Institutional Animal Care and Use Committee (IACUC) of National Chung Hsing University. IACUC No.: 103-131^R2^ (Valid From: 08/01/2015 to 07/31/2018). Eight weeks old female C57BL/6JNarl mice (*n* = 18) were obtained from the National Laboratory Animal Center, NLAC (Taipei, Taiwan). Mice were housed under specific pathogen-free, humidity and temperature controlled (20–25 °C) conditions and were fed AIN-93M diet (Dyets, Bethlehem, PA, USA). The lighting was operated on a 12 h light-dark cycle. After three weeks of acclimation, the mice were evenly divided into three groups (*n* = 6 per group) by body weight. Control animals received the same volume of saline by intraperitoneal injection (i.p.). MTX-treated mice received 0.3 mg MTX/kg body weight every other day [38] for 12 weeks. Mice in the MTX+FA group received the same dose of MTX, with a folinate (0.1 mg folinate/kg body weight) treatment 24 h [39] after the MTX injection. MTX was generally administered to RA patients at a dosage of 7.5–15 mg/wk [40,41,42]; folinate is generally given at the dosage of 2.5 mg/wk [43]. Doses used in the present study are [42] comparable to those in humans, receiving a clinically low-dose MTX and folinate rescue [30]. Based on the normal life expectancy of mice, the 12-wk MTX treatment may reflect a long-term (approximately 9 yrs) low-dose MTX therapy in humans.

In vivo kinetic studies were performed at the end of the study period. Mice were fed a modified L-amino acid defined diet with the same amino acid composition of the AIN-93M diet, except for the substitution of designated tracer: L-[2,3,3-^2^H_3_]serine (for 72 h) or [2-^13^C]glycine (for 144 h). Mice were sacrificed by isoflurane after an overnight fast. Plasma, liver, and bone marrow (washed by PBS) were stored in −80 °C until analysis.

### 4.6. Gas Chromatography/Mass Spectrometry Analysis

Cytosolic free amino acids and cellular protein were separately hydrolyzed, purified [44], and converted into heptafluorobutyryl n-propyl ester derivatives as described previously [45]. Isotopic enrichments in deoxyadenylate (dA) and deoxyguanylate (dG), and deoxythymidine (dT), and deoxymethyl-cytosine (mC) were determined in positive ionization mode by gas chromatography/mass spectrometry (GC/MS) as described previously [16,17,35]. Isotopic enrichments were determined in negative ionization for amino acids and in positive ionization mode for DNA using a model 6890 gas chromatograph and model 5973 mass spectrometer (Hewlett–Packard Corp., Palo Alto, CA, USA) as described previously [45].

### 4.7. Statistical Analysis

The differences among different treatment groups were examined by analysis of variance (ANOVA), then the comparisons of means between each two groups were determined using post-hoc analyses. For cell and animal data analyses, results are expressed as mean±SD. All statistical analyses were performed with SYSTAT 11.0 for Windows™ (Systat Software Inc., San Jose, CA, USA). For all analyses, the results were considered statistically significant if *p*-values were < 0.05.

## 5. Conclusions

In conclusion, we demonstrate novel findings that MTX specifically depletes mitochondrial 1C supply that can be ameliorated by folinate supplementation except for hepatic transmethylation. These results imply that clinical use of low-dose MTX may particularly impede 1C metabolism via depletion of mitochondrial formate. Formate supply and the methyl group homeostasis are potential targets for improving MTX therapy in the future. The MTX induced systematic and tissue-specific formate depletion needs to be addressed more carefully, and the efficacy of folinate with respect to protecting against such depletion deserves to be evaluated in medical practice.

## Figures and Tables

**Figure 1 ijms-22-01350-f001:**
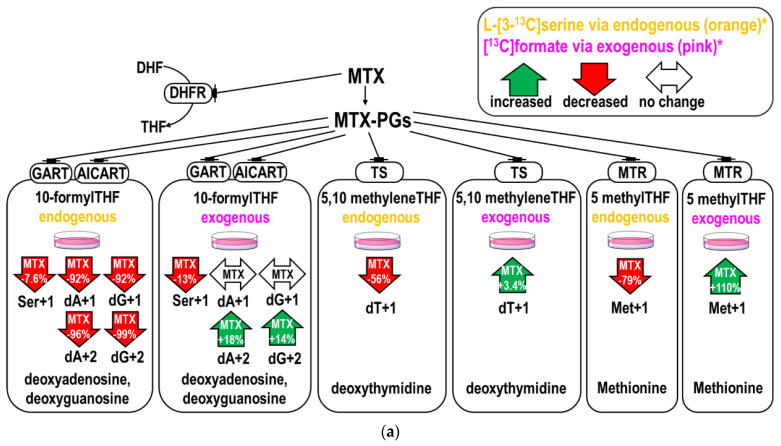
Effects of methotrexate (MTX) on endogenous and exogenous formate utilization. (**a**) The upper panel shows the percent changes of the enrichments in deoxyadenosine, deoxyguanosine, deoxythymidylate (dTMP), methionine, serine in cells treated with MTX. Endogenous and exogenous formate utilization were traced by L-[3-^13^C]serine (in orange) and exogenous [^13^C]formate (in pink). MTX decreased the incorporations of 1C moiety from L-[3-^13^C]serine (endogenous formate) and increased that from exogenous [^13^C]formate. The percent changes shown in the arrows indicate the isotopic enrichments in each target compound compared to controls. (**b**) The lower panel summarizes the metabolic fate of 1C flow from serine and formate. Solid arrows represent increased metabolic fluxes and dashed arrows represent decreased metabolic fluxes after MTX treatment.

**Figure 2 ijms-22-01350-f002:**
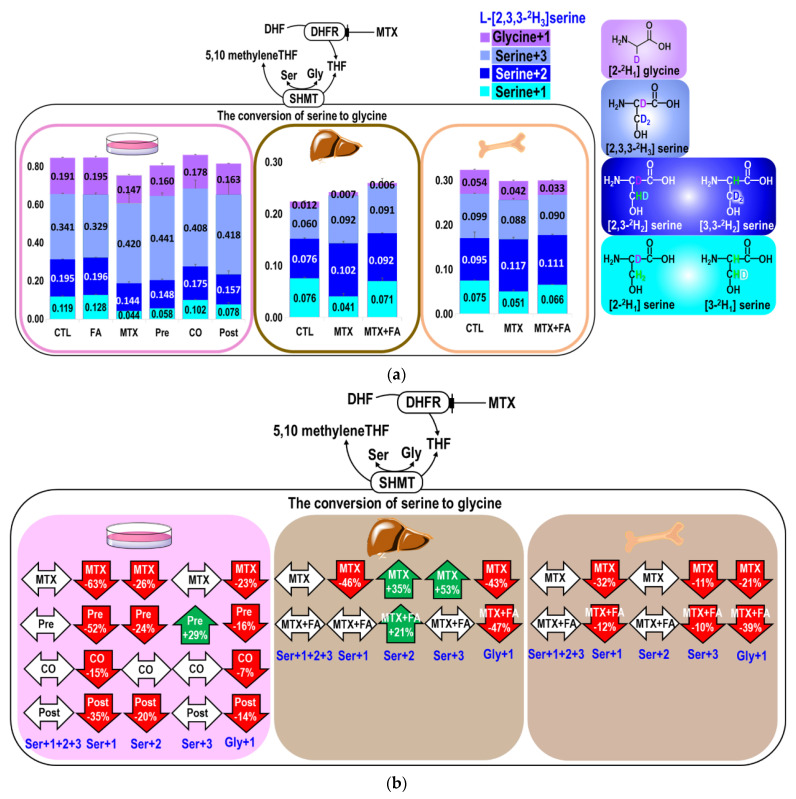
Effects of MTX on the conversion between serine and glycine in vitro and in vivo. (**a**) The upper panel shows the isotopic distributions of serine (Ser+1, +2, +3) and glycine (Gly+1) from tracer L-[2,3,3-^2^H_3_]serine in cell and mouse tissues treated with MTX and folinate. Enrichment of serine isomers (including serine+1 in cyan, serine+2 in blue, serine+3 in cornflower blue) and glycine (purple) were plotted as cumulative bar charts, under treatments of MTX and folinate in the cell model, mouse liver, and bone marrow. The isotopic distributions in glycine and serine isomers changed by MTX and folinate despite that the total of all serine isotopic species (Ser M+1+2+3) was nearly unchanged. Error bars represent the standard error of the mean. The chemical structure of isotopic serine isomers and glycine are illustrated on the right with the designated color. (**b**) The lower panel shows the percent changes in the isotopic enrichments in each target compound compared to the control cells/animals that received phosphate-buffered saline (PBS).

**Figure 3 ijms-22-01350-f003:**
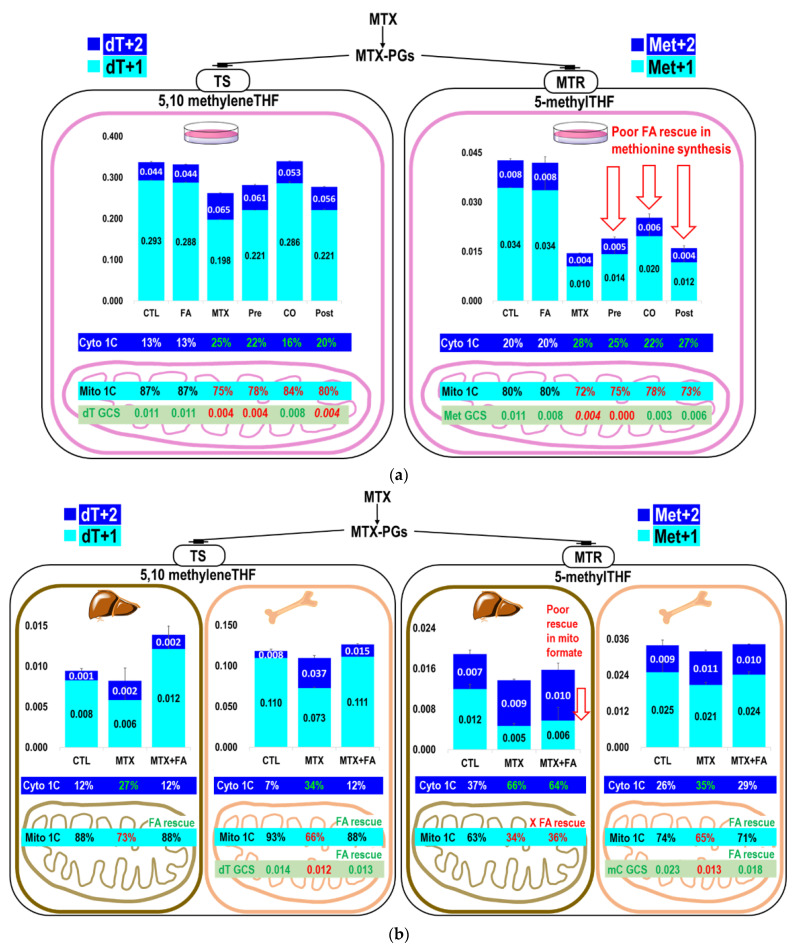
Effects of MTX on the partitioning between mitochondrial and cytosolic derived formate 1-carbon metabolic fluxes in vitro and in vivo. MTX and folinate alter the partitioning of 1C metabolic fluxes between mitochondrial (cyan M+1) and cytosolic (blue M+2) derived formate in dTMP and methionine synthesis in the (**a**) cell model and (**b**) mouse liver and bone marrow. Enrichment of dTMP (including dT+1 in cyan, dT+2 in blue) and methionine (including met+1 in cyan, met+2 in blue) were plotted as cumulative bar charts. Error bars represent the standard error of the mean. The proportional fluxes from mitochondria were calculated and shown in percentage (in cyan). The proportion of dTMP (dT+1) and methionine (Met+1) synthesized via mitochondrial 1C (cyan) and cytosolic 1C (blue) from L-[2,3,3-^2^H_3_]serine were calculated and shown in the lower panel of the Figure 3a,b. Enrichments in dTMP (dT), methionine (Met), and deoxymethylcytosine (mC) derived from [2-^13^C]glycine, reflecting the utilization of GCS-derived formate, are shown in green. (**c**) The percent changes of the isotopic enrichments in each target compound compared to the control cells/animals received PBS. (**d**) Summary of the impacts of MTX with or without accompanied folinate treatment on nucleotide and methionine syntheses between mitochondrial and cytosolic derived formate in vitro and in vivo. * denote the labeled metabolite from the designated tracer. Bold arrows represent increased metabolic fluxes; dashed arrows represent decreased metabolic fluxes after MTX treatment.

**Figure 4 ijms-22-01350-f004:**
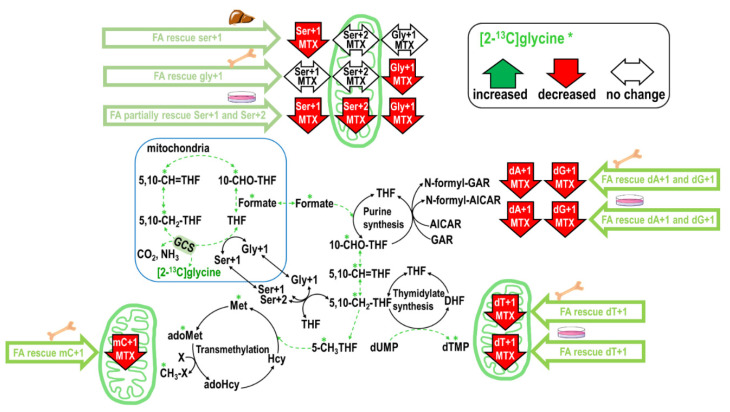
Summary of the impacts of MTX with or without accompanied folinate treatment on nucleotide and methionine syntheses from GCS in vitro and in vivo. * Denote the labeled metabolite from the designated tracer. Dashed arrows represent decreased metabolic fluxes after MTX treatment (in green).

**Figure 5 ijms-22-01350-f005:**
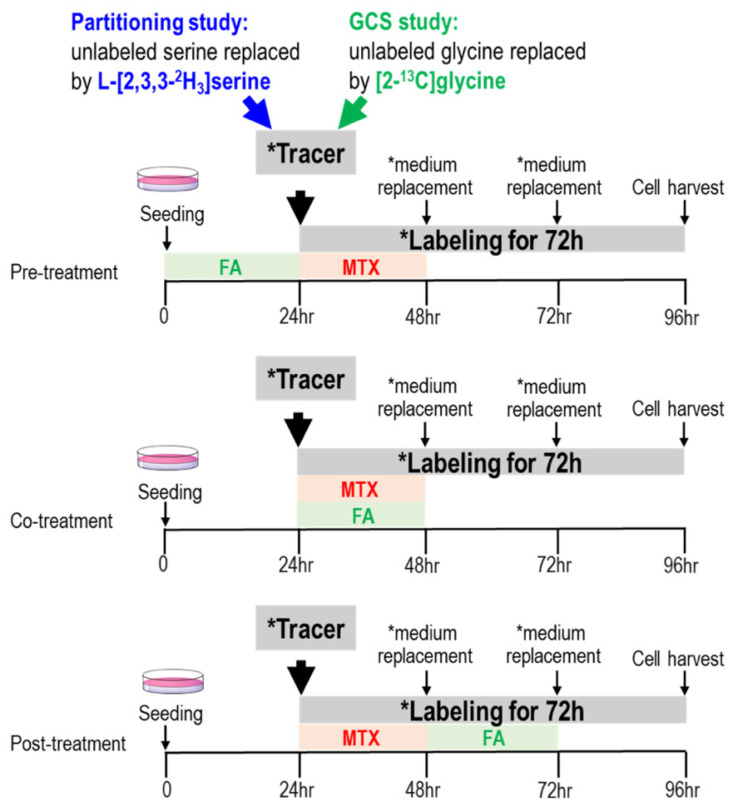
Tracing endogenous and exogenous formate utilization and the partitioning between mitochondrial and cytosolic derived formate. In vitro experimental design for investigating the efficacy of folinate rescue on MTX treatment. HepG2+GNMT cells were treated with MTX (50 nM) with and without folinate supplementation (100 nM) before, during, or after the 24 h MTX treatment, defined as “pre-treatment” (upper), “co-treatment” (middle), and “post-treatment” (lower), respectively. * Denotes medium with designated tracer.

**Figure 6 ijms-22-01350-f006:**
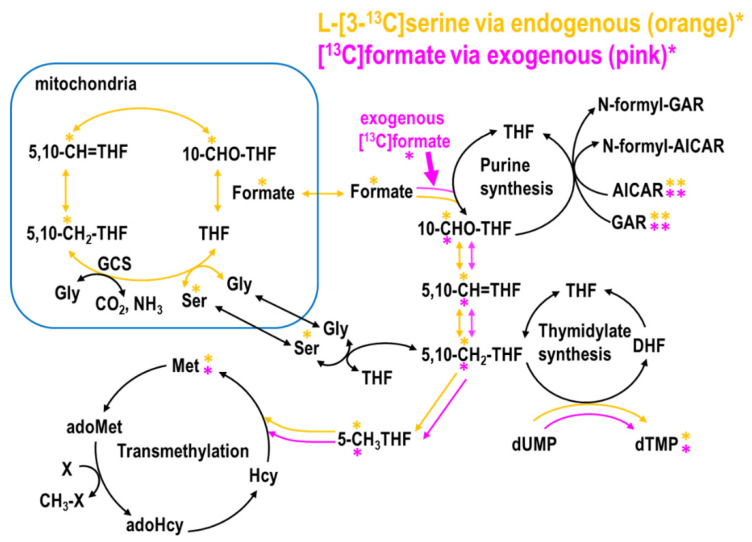
Tracing endogenous and exogenous formate utilization. Tracing 1C metabolic fluxes using L-[3-^13^C]serine and [^13^C]formate tracers entering purine, dTMP, and methionine syntheses via folate cofactor 10-CHO-THF, 5,10-CH_2_-THF, and 5-CH_3_THF. On the upper right, purine synthesis requires 10-CHO-THF by deoxyguanosine (GART) and aminoimidazole carboxamide ribonucleotide transformylase (AICART). On the lower right, 5,10-CH_2_-THF is used for dTMP synthesis. On the lower left, 5-methylTHF is used for methionine synthesis. Enrichments in the target metabolites were determined to investigate the impacts of MTX on the incorporations of one carbon moiety from endogenous formate (using L-[3-^13^C]serine, shown in orange, and exogenous [^13^C]formate, pink). * Denotes the labeled metabolite from the designated tracer.

**Figure 7 ijms-22-01350-f007:**
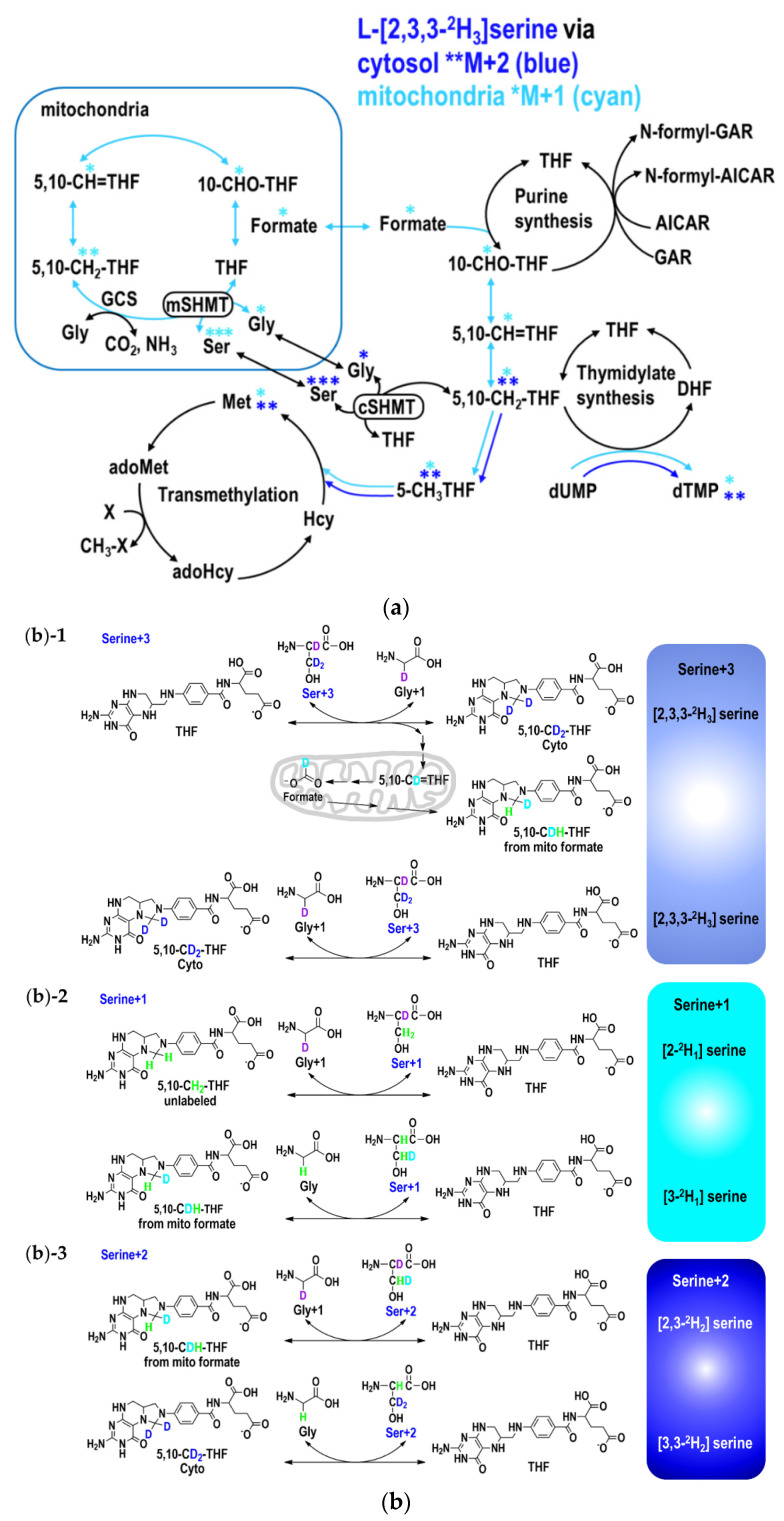
(**a**) Tracing the partitioning of 1 carbon flux between mitochondrial and cytosolic derived one carbon moiety via L-[2,3,3-^2^H_3_]serine. L-[2,3,3-^2^H_3_]serine tracer enables distinguishing mitochondrial (shown in cyan M+1) derived one carbon moiety from cytosolic (blue M+2) counterpart. On the lower right, generation of de novo [^2^H_1_]-thymidine from mitochondria-derived formate via mitochondrial serine hydroxymethyl-transferase (mSHMT) and de novo [^2^H_2_] thymidine from cytosol via cytosolic SHMT(cSHMT). On the lower left, generation of [^2^H_1_] methionine from mitochondrial-derived formate and [^2^H_2_] methionine from cytosol via cSHMT. Enrichments in target metabolites were determined to investigate the impacts of MTX and folinate on formate utilization for thymidine and methionine syntheses from mitochondria (cyan M+1) and cytosol (blue M+2). * Denotes the labeled metabolites. *, **, *** Denotes M+1, M+2, M+3, respectively. (**b**) Putative isotopic enrichments in serine (M+1, M+2, M+3) and glycine (M+1) from L-[2,3,3-^2^H_3_]serine tracer. (**b**)-1 Putative isotopic enrichment of serine+3: L-[2,3,3-^2^H_3_]serine (Ser+3) is labeled with one deuterium atom at the C-2 (purple) and another two at C-3 position (blue). The synthesis of 5,10-CH_2_THF acquires the methylene group (-CH_2_-) from serine, generating glycine. The deuterium atom in 5,10-CD_2_THF from cytosol via cSHMT and that from mitochondria-derived formate (CDOOH) are shown in blue and cyan, respectively. The unlabeled H is shown in green. In the reverse reaction, L-[2,3,3-^2^H_3_]serine (Ser+3) can be synthesized from [2-^2^H_1_] glycine with the incorporation of the labeled methylene group of 5,10-CD_2_THF from cytosol. (**b**)-2 Putative isotopic enrichment of serine+1: continued from (**b**)-1, but in the reverse reaction, [2-^2^H_1_] serine (Ser+1) is synthesized from [2-^2^H_1_] glycine with the incorporation of an unlabeled methylene group from 5,10-CH_2_THF, via cSHMT. Alternatively, [3-^2^H_1_]serine (Ser+1) can be synthesized from unlabeled glycine with the incorporation of the labeled methylene group of 5,10-CDHTHF from mitochondria-derived formate. (**b**)-3 Putative isotopic enrichment of serine+2: as for M+2 of serine (Ser+2), [2,3-^2^H_2_]serine can be synthesized from [2-^2^H_1_] glycine with the incorporation of labeled methylene group of 5,10-CDH-THF from mitochondria-derived formate. Or, [3,3-^2^H_2_]serine (Ser+2) can be synthesized from unlabeled glycine with the incorporation of labeled methylene group of 5,10-CD_2_THF from cytosol.

**Table 1 ijms-22-01350-t001:** Low-dose methotrexate (MTX) changes endogenous and exogenous formate utilization for nucleotide biosynthesis.

(**A**) Purine synthesis from L-[3-^13^C]serine derived formate and exogenous [^13^C]formate ^1,2^.						
	**Endogenous Formate from L-[3-^13^C]serine**			**Exogenous [^13^C]formate Incorporation**		
	**dA+1 ^3^**	**dA+2 ^3^**	**dA (MIA) ^4,6^**	**dA+1 ^3^**	**dA+2 ^3^**	**dA (MIA) ^4,6^**
**CTL ^4^**	0.310 ± 0.003	0.179 ± 0.003	1.155 ± 0.021	0.319 ± 0.002	0.121 ± 0.003	0.756 ± 0.025
**MTX ^4^**	**0.025 ± 0.014**	**0.008 ± 0.002**	**0.648 ± 0.154**	0.324 ± 0.005	**0.142 ±0.003**	**0.876 ± 0.007**
***p-value*^2^**	**0.001**	**0.000**	**0.044**	0.281	**0.022**	**0.023**
***%change*^5^**	**−92.06%**	**−95.54%**	**−50.73%**	1.65%	**+17.72%**	**+11.94%**
	**dG+1 ^3^**	**dG+2 ^3^**	**dG (MIA) ^4,6^**	**dG+1 ^3^**	**dG+2 ^3^**	**dG (MIA) ^4,6^**
**CTL ^4^**	0.321 ± 0.015	0.171 ± 0.007	1.066 ± 0.042	0.323 ± 0.008	0.119 ± 0.001	0.736 ± 0.013
**MTX ^4^**	**0.025 ± 0.010**	**0.001 ± 0.002**	**0.120 ± 0.169**	0.336 ± 0.006	**0.135 ± 0.002**	**0.802 ± 0.001**
***p-value*^2^**	**0.002**	**0.001**	**0.017**	0.199	**0.011**	**0.019**
***%change*^5^**	**−92.34%**	**−99.14%**	**−94.59%**	4.27%	**+13.70%**	**+6.64%**
(**B**) Relative incorporation of L-[3-^13^C]serine derived formate and exogenous [^13^C]formate in purines ^1,2^.						
	**Endogenous Formate from L-[3-^13^C]serine**			**Exogenous [^13^C]formate Incorporation**		
	**Ser+1 ^3^**	**dA+1 ^3^**	**dA+1/Ser+1 ^7^**	**Ser+1 ^3^**	**dA+1 ^3^**	
**CTL ^4^**	0.515 ± 0.016	0.310 ± 0.003	0.602 ± 0.006	0.189 ± 0.007	0.319 ± 0.002	
**MTX ^4^**	*0.475 ± 0.008*	**0.025 ± 0.014**	**0.052 ± 0.029**	**0.166 ± 0.000**	0.324 ± 0.005	
***p-value*^2^**	*0.087*	**0.001**	**0.001**	**0.046**	0.281	
***%change*^5^**	*−7.60%*	**−92.06%**	**−55.05%**	**−12.50%**	1.65%	
	**Ser+1 ^3^**	**dG+1 ^3^**	**dG+1/Ser+1 ^7^**	**Ser+1 ^3^**	**dG+1 ^3^**	
**CTL ^4^**	0.515 ± 0.016	0.321 ± 0.015	0.624 ± 0.029	0.189 ± 0.007	0.323 ± 0.008	
**MTX ^4^**	*0.475 ± 0.008*	**0.025 ± 0.010**	**0.052 ± 0.021**	**0.166 ± 0.000**	0.336 ± 0.006	
***p-value*^2^**	*0.087*	**0.002**	**0.002**	**0.046**	0.199	
***%change*^5^**	*−7.60%*	**−92.34%**	**−57.21%**	**−12.50%**	4.27%	
(**C**) Thymidine synthesis from L-[3-^13^C]serine derived formate and exogenous [^13^C]formate ^1,2^.						
	**Endogenous Formate from L-[3-^13^C]serine**			**Exogenous [^13^C]formate Incorporation**		
	**Ser+1 ^3^**	**dT+1 ^3^**	**dT+1/Ser+1 ^7^**	**Ser+1 ^3^**	**dT+1 ^3^**	
**CTL ^4^**	0.515 ± 0.016	0.452 ± 0.003	0.879 ± 0.006	0.189 ± 0.007	0.384 ± 0.002	
**MTX ^4^**	*0.475 ± 0.008*	**0.198 ± 0.022**	**0.417 ± 0.047**	**0.166 ± 0.000**	**0.397 ± 0.001**	
***t*-Test ^2^**	*0.087*	**0.004**	**0.005**	**0.046**	**0.017**	
**%change ^5^**	*−7.60%*	**−56.14%**	**−46.14%**	**−12.50%**	**+3.42%**	

^1^ To investigate formate utilization, L02 cells cultured in minimum essential medium (MEM) media were incubated with addition glycine, vitamin B12, and L-[3-^13^C]serine to reach a final concentration of 0.667 mM glycine, 0.001 mM B12, and 0.25 mM serine. In a parallel experiment, 0.25 mM [^13^C]formate was used instead of serine. ^2^ Enrichment of isotopic tracers with or without MTX treatment is expressed as mean ± SD (*n* = 2–3). Data are compared by student’s t-test. Bold values indicate statistically significant differences (*p* < 0.05). Italic values indicated a trend of difference (*p* < 0.05, *p* < 0.1). ^3^ dA+1: deoxyadenosine enrichments; dG+1: deoxyguanosine; dT+1: deoxythymidine; Ser+1: serine enrichments from L-[3-^13^C]serine or from [^13^C]formate. ^4^ CTL: control; MTX: methotrexate; MIA: mass isotopomer analysis. ^5^ The % change of the isotopic enrichments in MTX treated cells compared to control cells. ^6^ Determined from the ratio of the M+1 and M+2 isomers of dA and dG. A value of 1.0 would indicate that 100% of the C-2 and C-8 carbons of the purine ring were derived from [^13^C]formate. ^7^ The relative enrichment in deoxythymidine (dT+1) synthesis or purine (dA+1 or dG+1) from L-[3-^13^C]serine tracer was calculated and abbreviated as dT+1/Ser+1, dA+1/Ser+1, dG+1/Ser+1, respectively.

**Table 2 ijms-22-01350-t002:** Methotrexate (MTX) and folinate (FA) change (**B**,**D**) the partitioning of 5,10 methyleneTHF^8^ (5,10-CH_2_THF) 1C between methionine and thymidylate syntheses from endogenous and exogenous formate; and (**A**) relative incorporation of methylene group and (**C**) the partitioning of mitochondrial and cytosolic derived formate via 5,10-CH_2_THF.

(**A**) Relative incorporations of L-[3-^13^C]serine derived formate and exogenous [^13^C]formate in methionine ^1,2^.					
	**Endogenous Formate from L-[3-^13^C]serine**			**Exogenous [^13^C]formate Incorporation**	
	**Ser+1 ^6^**	**Met+1 ^6^**	**Met+1/Ser+1 ^7^**	**Ser+1 ^6^**	**Met+1 ^6^**
**CTL ^4^**	0.515 ± 0.016	0.014 ± 0.003	0.027 ± 0.006	0.189 ± 0.007	0.012 ± 0.004
**MTX ^4^**	*0.475 ± 0.008*	*0.003 ± 0.003*	*0.006 ± 0.007*	**0.166 ± 0.000**	**0.024 ± 0.001**
***t*-Test**	*0.087*	*0.074*	*0.083*	**0.046**	**0.038**
**%change ^5^**	*−7.60%*	*−78.90%*	*−2.1%*	**−12.50%**	**+110.00%**
(**B**) Partitioning of 5,10 methyleneTHF between methionine and thymidylate syntheses using endogenous formate (from L-[3-^13^C]serine) and exogenous [^13^C]formate in vitro ^1,2^.					
**L-[3-^13^C]serine**	**Ser+1 ^6^**	**dT+1 ^6^**	**Met+1 ^6^**	**dT+1/Met+1 ^8^**	
**CTL ^4^**	0.515 ± 0.016 ^a^	0.452 ± 0.003 ^a^	0.014 ± 0.003 ^a^	32.568 ± 0.208 ^a^	
**MTX ^4^**	*0.475 ± 0.008* ^b^	**0.198 ± 0.022 ^b^**	*0.003 ± 0.003 ^b^*	**67.634 ± 7.554 ^b^**	
**[^13^C]formate**	**Ser+1 ^6^**	**dT+1 ^6^**	**Met+1 ^6^**	**dT+1/Met+1 ^8^**	
**CTL ^4^**	0.189 ± 0.007 ^a^	0.384 ± 0.002 ^a^	0.012 ± 0.004 ^a^	33.410 ± 0.139 ^a^	
**MTX ^4^**	**0.166 ± 0.000 ^b^**	**0.397 ± 0.001 ^b^**	**0.024 ± 0.001 ^b^**	**16.450 ± 0.024 ^b^**	
(**C**) Relative incorporation of the methylene group from L-[2,3,3-^2^H_3_]serine into methionine and thymidylate syntheses with MTX and folinate rescue in vitro ^1,3^.					
**L-[2,3,3-^2^H_3_]serine**		**Via 5-methylTHF**		**Via 5,10 methyleneTHF**	
	**Ser+1+2+3 ^6^**	**Met+1+2 ^6^**	**Met from Ser ^7^**	**dT+1+2 ^6^**	**dT from Ser ^7^**
**CTL ^4^**	0.656 ± 0.001 ^a^	0.043 ± 0.000 ^a^	0.065 ± 0.000 ^a^	0.337 ± 0.003 ^a^	0.514 ± 0.006 ^a^
**FA ^4^**	0.653 ± 0.005 ^a^	0.042 ± 0.007 ^ac^	0.064 ± 0.011 ^ac^	0.331 ± 0.001 ^a^	0.507 ± 0.005 ^ac^
**MTX ^4^**	0.608 ± 0.041 ^a^	**0.015 ± 0.000 ^bc^**	**0.024 ± 0.002 ^b^**	**0.262 ± 0.002 ^b^**	**0.432 ± 0.027 ^bc^**
**MTX+FA Pre ^4^**	0.647 ± 0.010 ^a^	**0.019 ± 0.002 ^b^**	**0.029 ± 0.003 ^bc^**	**0.281 ± 0.002 ^c^**	**0.435 ± 0.004 ^b^**
**MTX+FA CO ^4^**	0.685 ± 0.037 ^a^	**0.025 ± 0.005 ^b^**	*0.037 ± 0.010* **^bc^**	0.339 ± 0.005 ^a^	0.496 ± 0.034 ^ab^
**MTX+FA Post ^4^**	0.653 ± 0.063 ^a^	**0.016 ± 0.005 ^b^**	**0.025 ± 0.010 ^b^**	0.277 ± 0.002 **^c^**	*0.426 ± 0.043* ^ab^
(**D**) Partitioning of 5,10 methyleneTHF 1C between methionine and thymidylate syntheses from L-[2,3,3-^2^H_3_]serine in vitro ^1,3^.					
	**From Mito**	**From Cyto**		**From Both**	
**L-[2,3,3-^2^H_3_]serine**	**dT+1/Met+1 ^8^**	**dT+2/Met+2 ^8^**		**dT+1+2/Met+1+2 ^8^**	
**CTL ^4^**	10.152 ± 0.103 ^a^	5.253 ± 0.182 ^a^		7.890 ± 0.148 ^a^	
**FA ^4^**	10.331 ± 1.643 ^a^	5.208 ± 0.096 ^a^		8.013 ± 1.340 ^ac^	
**MTX ^4^**	**17.220 ± 0.893 ^b^**	**16.032 ± 0.252 ^b^**		**18.082 ± 0.610 ^b^**	
**MTX+FA Pre ^4^**	*17.025 ± 2.420* ^b^	**12.669 ± 0.370 ^c^**		**14.956 ± 1.642 ^b^**	
**MTX+FA CO ^4^**	16.768 ± 3.576 ^ab^	**9.478 ± 0.161 ^d^**		*13.725 ± 2.733* ^bc^	
**MTX+FA Post ^4^**	19.548 ± 7.623 ^ab^	**13.162 ± 0.190 ^c^**		18.099 ± 5.308 ^ab^	

^1^ Enrichment of isotopic tracers with different combinations of folinate and MTX treatments is expressed as mean ± SD (*n* = 2–3). Data were calculated with a Student’s *t*-test. Values with different alphabetic superscripts (^a,b,c^) are significantly different (*p*-value < 0.05). Italic values indicated a trend of difference (*p* < 0.1) compared to the control group. ^2^ To investigate formate utilization, L02 cells cultured in MEM media were incubated with addition glycine, vitamin B12, and L-[3-^13^C]serine to reach a final concentration of 0.667mM glycine, 0.001 mM B12, and 0.25 mM serine. In a parallel experiment, 0.25 mM [^13^C]formate was used instead of serine. ^3^ HepG2 cells that stably express glycine-N methyltransferase (GNMT) to [19] were cultured in α-MEM medium and treated with 50 nM MTX and 100 nM folinate for 24 h. ^4^ CTL: control; MTX: methotrexate; FA: folinate supplementation; MTX+FA Pre: MTX with FA pre-treatment; MTX+FA CO: MTX with FA co-treatment; MTX+FA Post: MTX with FA post-treatment. ^5^ The % change of the isotopic enrichments in cells with specific treatment compared to control cells. ^6^ Ser+1: serine enrichments; dT+1: deoxythymidine; Met+1: methionine; Ser+1+2+3: the sum of all serine isotopic species; dT+1+2 and Met+1+2: deoxythymidine and methionine enrichment from both mitochondrial-derived formate and cytosolic CH_2_THF, respectively, from L-[2,3,3-^2^H_3_]serine tracer. ^7^ The relative enrichment in deoxythymidine (dT+1+2) and methionine (Met+1+2) synthesis from L-[2,3,3-^2^H_3_]serine tracer were calculated and abbreviated as dT+1+2/Ser+1+2+3 and Met+1+2/Ser+1+2+3, respectively. The relative enrichment in methionine (Met+1) synthesis from L-[3-^13^C]serine tracer was calculated and abbreviated as Met+1/Ser+1. ^8^ The partitioning of 5,10-CH_2_THF between methionine and dTMP was calculated as the ratio between the two and abbreviated as dT+1/Met+1, dT+2/Met+2 and dT+1+2/Met+1+2 from mitochondrial (mito) derived formate, cytosolic (cyto) CH_2_THF, and both, respectively.

**Table 3 ijms-22-01350-t003:** MTX and folinate change the utilization of serine in glycine synthesis in mouse bone marrow and liver proteins ^1^.

		Ser+1+2+3 ^2^	Ser+1 ^2^	Ser+2 ^2^	Ser+3 ^2^	Gly+1 ^2^
**Marrow**	**CTL ^3^**	0.270 ± 0.013 ^a^	0.075 ± 0.001 ^a^	0.095 ± 0.014 ^a^	0.099 ± 0.002 ^a^	0.054 ± 0.002 ^a^
	**MTX ^3^**	0.256 ± 0.010 ^a^	**0.051 ± 0.003 ^b^**	0.117 ± 0.004 ^a^	**0.088 ± 0.003 ^b^**	**0.042 ± 0.002 ^b^**
	**MTX+FA ^3^**	0.267 ± 0.007 ^a^	**0.066 ± 0.000 ^c^**	0.111 ± 0.002 ^a^	*0.090 ± 0.004* ^b^	**0.033 ± 0.001 ^c^**
**Liver**	**CTL ^3^**	0.212 ± 0.012 ^a^	0.076 ± 0.002 ^a^	0.076 ± 0.002 ^a^	0.060 ± 0.008 ^a^	0.012 ± 0.002 ^a^
	**MTX ^3^**	0.235 ± 0.017 ^a^	**0.041 ± 0.006 ^b^**	**0.102 ± 0.004 ^b^**	**0.092 ± 0.007 ^b^**	*0.007 ± 0.001* ^b^
	**MTX+FA ^3^**	0.254 ± 0.021 ^a^	0.071 ± 0.006 ^a^	**0.092 ± 0.001 ^b^**	0.091 ± 0.015 ^ab^	*0.006 ± 0.001* ^b^

^1^ Enrichment of isotopic tracer with or without the treatment of MTX or FA is expressed as mean ± SD (*n* = 2–3). Data were calculated with a Student’s *t*-test. Values with different alphabetic superscripts (^a,b,c^) were significantly different (*p*-value < 0.05). ^2^ Ser+1, Ser+2, Ser+3: enrichment of serine isomer distribution; Ser+1+2+3: the sum of all serine isotopic species; Gly+1: glycine enrichment; ^3^ CTL: control; MTX: methotrexate; MTX+FA: MTX with folinate rescue.

**Table 4 ijms-22-01350-t004:** MTX and folinate change the partitioning of methyleneTHF dependent 1C metabolic fluxes between mitochondrial and cytosolic derived formate in vivo.

(**A**) Relative incorporations of L-[2,3,3-^2^H_3_]serine in dTMP and methionine ^1^.						
		**via 5-methylTHF**		**via 5,10 methyleneTHF**		
		**Met+1+2 ^2^**	**Met from Ser ^4^**	**dT+1+2 ^2^**	**dT from Ser ^4^**	
**Marrow**	**CTL ^3^**	0.034 ± 0.005 ^a^	0.125 ± 0.012 ^a^	0.118 ± 0.003 ^a^	0.439 ± 0.031 ^a^	
	**MTX ^3^**	0.032 ± 0.001 ^a^	0.125 ± 0.003 ^a^	0.110 ± 0.004 ^a^	0.428 ± 0.002 ^a^	
	**MTX+FA ^3^**	0.034 ± 0.001 ^a^	0.128 ± 0.007 ^a^	0.126 ± 0.007 ^a^	0.472 ± 0.014 ^a^	
**Liver**	**CTL ^3^**	0.019 ± 0.002 ^a^	0.089 ± 0.013 ^a^	0.009 ± 0.001 ^a^	0.044 ± 0.002 ^a^	
	**MTX ^3^**	**0.014 ± 0.000 ^b^**	*0.058 ± 0.003 ^b^*	0.008 ± 0.003 ^a^	0.034 ± 0.011 ^a^	
	**MTX+FA ^3^**	0.016 ± 0.001 ^ab^	0.063 ± 0.010 ^ab^	0.014 ± 0.004 ^a^	0.054 ± 0.012 ^a^	
(**B**) Utilization of L-[2,3,3-^2^H_3_]serine for dTMP and methionine in vivo ^1^.						
	**via 5,10 methyleneTHF**			**via 5-methylTHF**		
	**dT+1 (Mito) ^2^**	**dT+2 (Cyto) ^2^**	**dT% ^5^**	**Met+1 (Mito) ^2^**	**Met+2 (Cyto) ^2^**	**met% ^5^**
	**Marrow**					
**CTL ^3^**	0.110 ± 0.005 ^a^	0.008 ± 0.002 ^a^	92.8 ± 2.3 ^a^	0.025 ± 0.003 ^a^	0.009 ± 0.002 ^a^	73.8 ± 1.5 ^a^
**MTX ^3^**	**0.073 ± 0.001 ^b^**	**0.037 ± 0.003 ^b^**	**66.2 ± 1.6 ^b^**	0.021 ± 0.001 ^a^	0.011 ± 0.000 ^a^	**64.8 ± 1.9 ^b^**
**MTX+FA ^3^**	0.111 ± 0.008 ^a^	*0.015 ± 0.001 ^c^*	88.1 ± 1.4 ^a^	0.024 ± 0.001 ^a^	0.010 ± 0.000 ^a^	70.5 ± 0.7 ^a^
	**Liver**					
**CTL ^3^**	0.008 ± 0.001 ^a^	0.001 ± 0.000 ^a^	87.9 ± 1.9 ^a^	0.012 ± 0.001 ^a^	0.007 ± 0.001 ^a^	63.4 ± 0.9 ^a^
**MTX ^3^**	*0.006 ± 0.002*^a^*	0.002 ± 0.002 ^a^	*72.7 ± 8.4*^a^*	**0.005 ± 0.000 ^b^**	*0.009 ± 0.000 ^b^*	**34.3 ± 2.8 ^b^**
**MTX+FA ^3^**	*0.012 ± 0.003*^a^*	0.002 ± 0.001 ^a^	88.3 ± 6.4 ^a^	*0.006 ± 0.003 ^b^*	*0.010 ± 0.001 ^b^*	*36.0 ± 13.4* ^b^
(**C**) Partitioning of 5,10 methyleneTHF in methionine and dTMP syntheses from L-[2,3,3-^2^H_3_]serine in vivo ^1^.						
		**Mito**	**Cyto**		**Total**	
		**dT+1/Met+1 ^6^**	**dT+2/Met+2 ^6^**		**dT+1+2/Met+1+2 ^6^**	
**Marrow**	**CTL ^3^**	4.390 ± 0.201 ^a^	0.953 ± 0.278 ^a^		3.486 ± 0.075 ^a^	
	**MTX ^3^**	**3.507 ± 0.046 ^b^**	**3.301 ± 0.278 ^b^**		3.435 ± 0.128 ^a^	
	**MTX+FA ^3^**	4.601 ± 0.326 ^a^	1.489 ± 0.092 ^a^		3.684 ± 0.203 ^a^	
**Liver**	**CTL ^3^**	0.693 ± 0.051 ^a^	0.166 ± 0.042 ^a^		0.500 ± 0.048 ^a^	
	**MTX ^3^**	1.239 ± 0.360 ^ab^	0.264 ± 0.176 ^a^		0.598 ± 0.239 ^a^	
	**MTX+FA ^3^**	**2.111 ± 0.487 ^b^**	0.176 ± 0.137 ^a^		0.882 ± 0.265 ^a^	

^1^ Enrichment of isotopic tracer with or without the treatment of MTX or FA is expressed as mean ± SD (*n* = 2–3). Data were calculated with a Student’s *t*-test. Values with different alphabetic superscripts (^a,b,c^) were significantly different (*p*-value < 0.05). Italic values with * indicated a trend of difference (* *p* < 0.1) compared to the control group. ^2^ dT+1, dT+2, and dT+1+2: deoxythymidine enrichment from mitochondrial-derived formate, cytosolic CH_2_THF, and both, respectively. Met+1, Met+2, and Met+1+2: methionine enrichment from mitochondrial-derived formate, cytosolic CH_2_THF, and both, respectively. ^3^ CTL: control; MTX: methotrexate; MTX+FA: MTX with folinate rescue. ^4^ The relative enrichment in deoxythymidine (dT+1+2) and methionine (Met+1+2) synthesis from L-[2,3,3-^2^H_3_]serine tracer were calculated and abbreviated as dT+1+2/Ser+1+2+3 and Met+1+2/Ser+1+2+3, respectively. ^5^ The ratio of deoxythymidine (dT+1) and methionine (Met+1) synthesized via mitochondrial-derived formate from L-[2,3,3-^2^H_3_]serine were calculated and abbreviated as dT+1/dT+1+2 and Met+1/Met+1+2, respectively; ^6^ The ratio of deoxythymidine synthesis to methionine synthesis was calculated and abbreviated as dT+1/Met+1 (from mitochondrial-derived formate), dT+2/Met+2 (from cytosolic CH_2_THF), and dT+1+2/Met+1+2 (from both mitochondrial-derived formate and cytosolic CH_2_THF).

**Table 5 ijms-22-01350-t005:** Methotrexate (MTX) significantly inhibits formate production via mitochondrial glycine cleavage system (GCS) that can be rescued by folinate in vitro and in vivo.

(**A**) Nucleotide synthesis from [2-^13^C]glycine in vitro ^1,2^.							
**[2-^13^C]glycine**	**Ser+1 ^3^**	**Gly+1 ^3^**	**dA+1 ^3^**	**dG+1 ^3^**	**dT+1 ^3^**	**Ser+2 ^3^**	
**CTL ^4^**	0.262 ±0.006 ^a^	0.329 ± 0.005 ^a^	0.176 ± 0.001 ^a^	0.220 ± 0.004 ^a^	0.011 ± 0.001 ^a^	0.008 ± 0.001 ^a^	
**FA ^4^**	0.268 ± 0.006 ^a^	0.278 ± 0.034 ^a b^	0.175 ± 0.002 ^a^	0.212 ± 0.008 ^ad^	0.011 ± 0.000 ^a^	0.008 ± 0.002 ^a^	
**MTX ^4^**	**0.151 ± 0.001 ^b^**	**0.269 ± 0.009 ^b^**	**0.095 ± 0.001 ^b^**	**0.142 ± 0.000 ^b^**	**0.004 ± 0.002 ^b c^**	**0.000 ± 0.000 ^b^**	
**MTX+FA Pre ^4^**	**0.164 ± 0.000 ^c^**	**0.160 ±0.032 ^c^**	**0.112 ± 0.002 ^c^**	**0.156 ± 0.003 ^c^**	**0.004 ± 0.000 ^c^**	0.006 ± 0.006 ^a b c^	
**MTX+FA CO ^4^**	**0.222 ± 0.006^d^**	0.248 ± 0.070 ^a b c^	**0.153 ± 0.001^d^**	**0.201 ± 0.001^d^**	0.008 ± 0.002 ^a b^	**0.003 ± 0.001 ^c^**	
**MTX+FA Post ^4^**	**0.179 ± 0.000 ^e^**	**0.265 ± 0.010 ^b^**	**0.084 ± 0.002 ^e^**	**0.126 ± 0.001 ^e^**	*0.004 ± 0.003 ^b c^*	**0.000 ± 0.000 ^b^**	
(**B**) Utilization of glycine for serine synthesis in bone marrow and liver ^1^.							
		**In Protein Lysates**		**In Cytosol Free Amino Acid**			
		**Ser+1 ^3^**	**Gly+1 ^3^**		**Ser+1 ^3^**	**Gly+1 ^3^**	
**Marrow**	**CTL ^4^**	0.068 ± 0.001 ^a^	0.057 ± 0.001 ^a^		0.070 ± 0.004 ^a^	0.068 ± 0.002 ^a^	
	**MTX ^4^**	0.075 ± 0.005 ^a b^	**0.038 ± 0.006 ^b^**		0.069 ± 0.003 ^a^	0.070 ± 0.003 ^a^	
	**MTX+FA ^4^**	**0.084 ± 0.002 ^b^**	**0.073 ± 0.000 ^c^**		0.074 ± 0.003 ^a^	*0.061 ± 0.003 ^b^*	
**Liver**	**CTL ^4^**	0.125 ± 0.001 ^a^	0.020 ± 0.001 ^a^		0.097 ± 0.009 ^a^	0.111 ± 0.003 ^a^	
	**MTX ^4^**	*0.115 ± 0.004 ^b^*	0.061 ± 0.042 ^a^		0.102 ± 0.004 ^a^	0.110 ± 0.014 ^a b^	
	**MTX+FA ^4^**	0.125 ± 0.002 ^a b^	0.032 ± 0.035 ^a^		0.105 ± 0.003 ^a^	**0.127 ± 0.002 ^b^**	
(**C**) Nucleotide synthesis from [2-^13^C]glycine in bone marrow ^1^.							
	**Ser+1 ^3^**	**Gly+1 ^3^**	**dT+1 ^3^**	**mC+1 ^3^**	**Ser+2 ^3^**	**dA+1 ^3^**	**dG+1 ^3^**
**CTL ^4^**	0.068 ± 0.001 ^a^	0.057 ± 0.001 ^a^	0.014 ± 0.000 ^a^	0.023 ± 0.002 ^a^	0.0013 ± 0.000 ^a^	0.112 ± 0.003 ^a^	0.113 ± 0.001 ^a^
**MTX ^4^**	0.075 ± 0.005 ^a b^	**0.038 ± 0.006 ^b^**	**0.012 ± 0.000 ^b^**	**0.013 ± 0.000 ^b^**	0.0008 ± 0.001 ^a^	**0.097 ± 0.003 ^b^**	**0.095 ± 0.002 ^b^**
**MTX+FA ^4^**	**0.084 ± 0.002 ^b^**	**0.073 ± 0.000 ^c^**	0.013 ± 0.001 ^a^	0.018 ± 0.002 ^a^	0.0019 ± 0.000 ^a^	0.107 ± 0.003 ^a^	0.105 ± 0.005 ^a b^

^1^ Enrichment of isotopic tracers with varied treatment is expressed as mean ± SD (*n* = 2–3). Data were calculated with a Student’s *t*-test. Values with different alphabetic superscripts (^a^,^b^,^c^) are significantly different (*p*-value < 0.05). Italic values indicated a trend of difference (*p* < 0.1) compared to the control group. ^2^ HepG2 cells that stably express GNMT to [19] were cultured in α-MEM medium and treated with 50 nM MTX and 100 nM folinate for 24 hr. ^3^ dA+1: deoxyadenosine enrichments; dG+1: deoxyguanosine; dT+1: deoxythymidine; Ser+1: serine; Gly+1: glycine enrichment from [2-^13^C]glycine; Ser+2: serine enrichment via mitochondria glycine cleavage system; mC+1: 5-methylcytosine enrichment. ^4^ CTL: control; MTX: methotrexate; FA: folinate supplementation; MTX+FA Pre: MTX with FA pre-treatment; MTX+FA CO: MTX with FA co-treatment; MTX+FA Post: MTX with FA post-treatment; MTX+FA: MTX with folinate rescue.

## Data Availability

The data presented in this study are available on request from the corresponding author.

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
