# Peer review of "Folinate Supplementation Ameliorates Methotrexate Induced Mitochondrial Formate Depletion In Vitro and In Vivo"

_ijms, 2021, doi:10.3390/ijms22031350_

Round 1

Reviewer 1 Report

The abstract contains acronym “DMARD”. Please write the full form.

Please correct the spelling in the following sentence:

“Methods: Combing animal 19 model (8-week old female C57BL/6JNarl mice, n=18)”,

The study involves the use of female mice, is there any reason for the this? Why were male mice not involved?

How was the weight and food intake affected by different treatments?

The authors mention that” long-term clinical MTX treatment in humans may lead to systematic and tissue  specific formate depletion that needs to be addressed more carefully”

What is the recommendations according to the authors?

Author Response

Please see attached reply letter

Reviewer 2 Report

Chiang and coworkers evaluated the effect of folinate supplementation for methotrexate-induced mitochondrial formate depletion. The study is well designed from multiple aspects, covering comprehensive datasets from in vitro to in vivo. The conclusion that low-dose MTX may impede 1C metabolism via depletion of mitochondrial formate is of clinical importance. Therefore, this work is recommended for publication. 

Author Response

please see attached reply letter
